# Cruciate Ligament Cell Sheets Can Be Rapidly Produced on Thermoresponsive poly(glycidyl ether) Coating and Successfully Used for Colonization of Embroidered Scaffolds

**DOI:** 10.3390/cells10040877

**Published:** 2021-04-12

**Authors:** Ingrid Zahn, Daniel David Stöbener, Marie Weinhart, Clemens Gögele, Annette Breier, Judith Hahn, Michaela Schröpfer, Michael Meyer, Gundula Schulze-Tanzil

**Affiliations:** 1Institute of Anatomy and Cell Biology, Paracelsus Medical University, Nuremberg and Salzburg, Prof. Ernst Nathan Str. 1, 90419 Nuremberg, Germany; ingrid.zahn@fau.de (I.Z.); clemens.goegele@pmu.ac.at (C.G.); 2Department of Applied Chemistry, Nuremberg Institute of Technology Georg Simon Ohm, Keßlerplatz 12, 90489 Nuremberg, Germany; 3Institute of Functional and Clinical Anatomy, Friedrich Alexander University, Erlangen-Nuremberg, Universitätsstr. 19, 91054 Erlangen, Germany; 4Institute of Chemistry and Biochemistry, Freie Universität Berlin, Takustrasse 3, 14195 Berlin, Germany; daniel.stoebener@fu-berlin.de (D.D.S.); marie.weinhart@fu-berlin.de (M.W.); 5Institute of Physical Chemistry and Electrochemistry, Leibniz Universität Hannover, Callinstr. 3A, 30167 Hannover, Germany; 6Department of Biosciences, Paris Lodron University Salzburg, Hellbrunnerstraße 34, 5020 Salzburg, Austria; 7Leibniz-Institut für Polymerforschung Dresden e. V. (IPF), Hohe Straße 6, 01069 Dresden, Germany; breier@ipfdd.de (A.B.); hahn-judith@ipfdd.de (J.H.); 8FILK Freiberg Institute (FILK), Meißner Ring 1-5, 09599 Freiberg, Germany; michaela.schroepfer@filkfreiberg.de (M.S.); michael.meyer@filkfreiberg.de (M.M.)

**Keywords:** anterior cruciate ligament, ACL, cell sheet, thermoresponsive polymer, embroidered scaffolds, ligament tissue engineering

## Abstract

Anterior cruciate ligament (ACL) cell sheets combined with biomechanically competent scaffolds might facilitate ACL tissue engineering. Since thermoresponsive polymers allow a rapid enzyme-free detachment of cell sheets, we evaluated the applicability of a thermoresponsive poly(glycidyl ether) (PGE) coating for cruciate ligamentocyte sheet formation and its influence on ligamentocyte phenotype during sheet-mediated colonization of embroidered scaffolds. Ligamentocytes were seeded on surfaces either coated with PGE or without coating. Detached ligamentocyte sheets were cultured separately or wrapped around an embroidered scaffold made of polylactide acid (PLA) and poly(lactic-co-ε-caprolactone) (P(LA-CL)) threads functionalized by gas-phase fluorination and with collagen foam. Ligamentocyte viability, protein and gene expression were determined in sheets detached from surfaces with or without PGE coating, scaffolds seeded with sheets from PGE-coated plates and the respective monolayers. Stable and vital ligamentocyte sheets could be produced within 24 h with both surfaces, but more rapidly with PGE coating. PGE did not affect ligamentocyte phenotype. Scaffolds could be colonized with sheets associated with high cell survival, stable gene expression of ligament-related type I collagen, decorin, tenascin C and Mohawk after 14 d and extracellular matrix (ECM) deposition. PGE coating facilitates ligamentocyte sheet formation, and sheets colonizing the scaffolds displayed a ligament-related phenotype.

## 1. Introduction

Cell sheets represent highly confluent monolayers, associated with their self-made extracellular matrix (ECM). This allows an intimate cell–ECM interaction, which could stabilize the tissue-specific phenotype of cells and avoid dedifferentiation, which is often observed in monolayer cultures of primary cells, including tenocytes [1].

Cell sheets for tendon and ligament reconstruction are generally made from various stem cells [2,3,4,5] (Table 1). Cell sheets have been used for ligament tissue engineering; however, the production of cell sheets manufactured directly from anterior cruciate ligament (ACL)-derived primary ligamentocytes has barely been reported yet [6]. Cell sheets can be combined with scaffolds [4,6,7,8] and might allow a directed seeding of distinct scaffold areas with a particular cell type since they can be placed at defined areas of a scaffold [9]. Sheets produced with different or similar cell types can also be stacked [6,10]. This is of advantage for tissue engineering of zonal and multiple cell types containing tissues such as periodontal tissue [10] or the attachment part of the ACL to bone representing the so called enthesis. Cell sheet formation and maturation often require many weeks and sheet detachment is time-consuming and associated with the risk of cell layer damage [2]. 

Thermoresponsive coatings, which can switch by a thermal trigger from a cell-adhesive to a cell-repellent state, could help to gain much faster access to stable cell sheets, e.g., within 17 hours as demonstrated with stem cells [5].

These coatings are usually based on polymers, which undergo a thermally induced reversible phase transition in aqueous solution [11,12], as well as on surfaces at a certain temperature with a concomitant change in polymer chains’ hydration and conformation and thus result in reversibly switchable properties of the surface [13]. For cell culture applications the phase transition regime of the thermoresponsive coating on the culture dish needs to be located within the physiologically relevant temperature range below 37 °C. Hence, under standard cell culture conditions the coating is in its dehydrated, collapsed state, which ideally promotes adhesion and proliferation of anchorage-dependent mammalian cells for convenient culture, similar to standard tissue culture polystyrene dishes. Lowering the temperature to ambient temperature (AT), e.g., by simply taking the coated culture dishes out of the incubator, induces the swelling and rehydration of the coating, allowing an enzyme-free release of subconfluent single cells and confluent cell sheets [14,15]. In 1990, the first thermoresponsive coating based on poly(*N*-isopropylacrylamide) (PNIPAm) was developed on standard tissue culture polystyrene for the production of cell sheets [16,17], which is now commercially available under the tradename UpCell^™^ [18,19]. These PNIPAm-coated dishes were applied to harvest sheets of different cell types and have already been used for ligament sheet formation by ACL-derived CD34 positive stem cells [5], ACL fibroblasts and rotator cuff-derived cells [6,20] (Table 1). The intimate adaption of cell culture conditions including cell density and supplements are important for successful cell sheet production [14,21] (Table 1).

A novel thermoresponsive and cell-compatible coating was developed on the basis of glycidyl methyl ether (GME) and ethyl glycidyl ether (EGE) by Weinhart et al., representing low molecular weight (~2–3 kDa) poly(glycidyl ether) (PGE) copolymers with a GME:EGE-ratio of 1:3 on gold model substrates [22]. Essentially, glycidyl ether-based polymers such as thermoresponsive random PGE copolymers [23] represent a class of polymers with a high number of functional groups in the side chains of the polyether backbone for tailored modification according to the desired application. Hence, they have a high potential for tissue engineering due to low intrinsic toxicity, adjustable water solubility and high chain flexibility [24,25,26]. After basic structure–property relationship studies with PGEs on gold model substrates [27,28], these functional coatings for cell sheet fabrication were transferred to more commonly applied glass [29] and polystyrene cell culture dishes [14,30], which were used in this study to prepare cell sheets with lapine cruciate ligamentocytes. These ligamentocyte sheets were seeded on a biomechanically competent embroidered scaffold for ligamentogenesis. Since it has not been studied yet, whether the expression profile of ligamentocytes or respective stem/progenitor cells is affected by thermoresponsive surface coatings used for tendon/ligament cell sheet preparation (Table 1) was a particular focus in the present study.

Hence, key questions in this study are whether PGE allows the formation of sheets of primary cruciate ligamentocytes, whether the coating affects viability and ligamentocyte phenotype. Finally, it should be tested whether ligamentocyte sheets from PGE-coated surfaces can be used to seed embroidered scaffolds and if the ligament-related phenotype is then maintained on these scaffolds.

**Table 1 cells-10-00877-t001:** Sheet preparation techniques used for tendon and ligament tissue engineering.

Cell Type	Surface Used	Supplements	Time	Cell Density	Reference
LSPC (lapine ACL)	normal cell culture surface	50 ng/mL of ascorbic acid	2 w	6.00 × 10^3^ cells cm^−2^	[4]
MSCs (cell line) and TSPCs (human bone marrow, Achilles tendon)	normal cell culture surface	50 µg/mL ascorbic acid, high glucose conditions, 1 w	until confluence, >1 w	not provided	[31]
TDSC (human Achilles tendon)	normal cell culture surface	50 µg/mL ascorbic acid, high glucose conditions, 14 or 16.5 d	until confluence, >14 w	8.00 × 10^3^ cells cm^−2^	[32]
TDSC (rat patellar tendon)	normal cell culture surface	25 µM ascorbic acid (=4.4 µg/mL), 25 ng/mL CTGF	2 w	5.00 × 10^3^ cells cm^−2^	[33]
TDSC (rat patellar tendon)	normal cell culture surface	25 ng/mL CTGF 25 µM ascorbic acid (=4.4 µg/mL)	until confluence (~2 w)	6.00 × 10^3^ cells cm^−2^	[3]
ASC (canine)	collagen layer in insert	no	3–4 d	1.00 × 10^4^ cells cm^−2^	[34]
ASCs (human)	magnet array plate	iron oxide chitosan nanoparticles	7 d	6.25 × 10^4^ cells cm^−2^	[35]
MSCs (sheep, bone-marrow derived)	polycaprolactone (PCL) electrospun mesh	50 µg/mL ascorbic acid	4 w	5.97 × 10^3^ cells cm^−2^	[2]
ACL-derived CD34+ SCs (human)	UpCell (CellSeed: PNIPAm)	BMP2 overexpression	17 h	1.43 × 10^5^ cells cm^−2^	[5,36]
ACL fibroblasts (human)	UpCell (CellSeed: PNIPAm)	no	not mentioned	3.00 × 10^4^ cells cm^−2^	[6]
rotator cuff-derived cells (human supraspinatus tendon)	UpCell (CellSeed: PNIPAm)	no	17 h	2.63 × 10^5^ cells cm^−2^	[20]
TDSC (rat AS tendon)	UpCell (CellSeed: PNIPAm)	25 mM ascorbic acid (=4.4 µg/mL)	3 d	1.56 × 10^4^ cells cm^−2^	[37]
CL fibroblasts (lapine)	PGE	no	24 h	5–8 × 10^5^ cells cm^−2^	present study

ACL: anterior cruciate ligament, ASC: adipose tissue derived stem cells, BMP2: bone morphogenetic protein, CL: cruciate ligament, CTGF: connective tissue growth factor, d: day, h: hour, LSPC: ligament stem/progenitor cells, MSC: mesenchymal stromal cells, PCL: polycaprolactone, PGE: poly(glycidyl ether), SCs: stem cells, TDSC: tendon-derived stem cells, TNMD: tenomodulin, TSPCs: tendon specific progenitor cells, w: weeks.

## 2. Materials and Methods

### 2.1. Thermoresponsive Surface Coating with PGE on Polystyrene Plates

Thermoresponsive brushes based on glycidyl methyl ether (GME)/ethyl glycidyl ether (EGE) (1:3) copolymers were self-assembled onto polystyrene 12-well culture plates (Greiner, ThermoFisher Scientific Inc., Darmstadt, Germany) via the physical adsorption of a hydrophobic, photo-reactive benzophenone anchor block based on the monomer 4-[2-(2,3-epoxypropoxy)ethoxy]benzophenone (EEBP) as described in detail by Stöbener et al. [15]. Subsequently, the PGE brush layers were covalently immobilized onto the polystyrene surfaces by irradiation with ultraviolet light.

### 2.2. Preparation of Embroidered P(LA-CL)/PLA Scaffolds

Embroidered scaffolds (15 mm × 4 mm × 1 mm) were prepared using two different thread materials. The first one was a monofilament thread consisting of P(LA-CL), which was commercially produced (USP 7-0, Gunze Ldt., Osaka, Japan). The second material was a PLA multifilament (six filaments, Tt = 155 dtex, based on Ingeo biopolymer 6202D, (NatureWorks, Minnetonka, MN, USA)), which was melt spun at the Leibniz-Institut für Polymerforschung Dresden e. V. (IPF) in Dresden (Germany). Scaffolds were embroidered using an embroidery machine (JCZ 0209-550, ZSK Stickmaschinen GmbH, Krefeld, Germany). P(LA-CL) served as upper and PLA as lower thread in each ply for scaffold production on a water-soluble non-woven sheet of polyvinyl alcohol (PVA, Freudenberg Einlagestoffe KG, Weinheim, Germany). A zig-zag pattern design with 1.8 mm stitch length, 15° stitch angle and 0.2 mm duplication shift was used for embroidering. Three plies were stacked and locked together to design a three-dimensional (3D) scaffold. The PVA was washed out three times for 30 min in tap water on a compact shaker (KS 15 A, Edmund Bühler GmbH, Bodelshausen, Germany). The 3D scaffolds were dried at AT.

#### Functionalization of Embroidered P(LA-CL)/PLA Scaffolds

Embroidered scaffolds were fluorinated for 60 s in a fluorination batch reactor (Fluor-Technik-System GmbH, Lauterbach, Germany) at FILK (FILK Freiberg Institute gGmbH, Germany) using a mixture of 10% fluorine gas in air. After fluorination, scaffolds were flushed with synthetic air and then functionalized with purified native bovine type I collagen forming a refibrillated hydrogel, which is then lyophilized on the polymer threads of the embroidered polymer prepared at FILK. For stabilization the collagen was cross-linked with the gas phase of hexamethylene diisocyanate (HMDI, Merck KGaA, Darmstadt, Germany) in an exsiccator [38]. Scaffolds were sterilized by incubation in 70% ethanol (ETOH) for 30 min followed by rinsing three times in distilled water before incubation in fetal bovine serum (FBS, Bio&SELL, GmbH, Feucht, Germany) until seeding.

### 2.3. Isolation of Lapine Cruciate Ligamentocytes

Lapine cruciate ligaments used for cell isolation were obtained from the abattoir. Lapine cruciate ligamentocytes were isolated from four healthy female and one male New Zealand Rabbits (mean age of 12 months). Surrounding connective tissue was removed and 1–2-mm sized pieces of the cruciate ligaments were prepared and transferred into a T-25 culture flask (Sarstedt AG & Co. KG, Nürnbrecht, Germany) with 2 mL culture medium (Dulbecco’s Modified Eagle’s Medium (DMEM)/Ham’s F12 medium [1:1] (Bio&SELL), containing 10% fetal bovine serum (FBS, Bio&SELL), 1% penicillin/streptomycin solution, 25 μg/mL ascorbic acid (Sigma-Aldrich, Munich, Germany), 2.5 μg/mL amphotericin B (Bio&SELL) and MEM amino acid solution (Sigma-Aldrich)). Culture medium was changed every 2–3 days. After around 1 week, cruciate ligamentocytes emigrated from the explant. Cells were detached after being 80% confluent using 0.05% trypsin/0.02% ethylenediaminetetraacetic acid (EDTA) solution (Bio&SELL). Cell numbers and viability were calculated with a hemocytometer using the trypan blue exclusion assay.

### 2.4. Preparation of Ligamentocyte Sheets and Scaffold Seeding with Sheets

A total of 5–8 × 10^5^ ligamentocytes per cm^2^ from 3–5 different donors were seeded on a 12-well polystyrene cell culture plate coated with PGE for 24 h. As controls served cells, which were seeded into cell culture plates without PGE coating at identical cell density and cells cultured as monolayers (2.4 × 10^3^ per cm^2^) in a density normally used for cell expansion. After detachment with phosphate buffered saline (PBS, Bio&SELL) with Ca^++^ and Mg^++^ using a temperature shift between AT (21 °C, 10 min) and 37 °C (5 min), the cell sheets were cultivated for 7 d in a 24-well ultra-low attachment plate (CORNING, Tewksbury, MA, USA). Cell sheets from control plates were detached in a similar manner. If no sheet detachment was observed at 37 °C in the incubator, the plate was transferred to AT conditions again until sheet release was achieved. Sheet detachment was supported by pipetting PBS under the margins of the sheets in the well.

Since it was much less time-consuming to harvest intact ligamentocyte sheets from the PGE-coated surface, and cell sheet detachment from uncoated surfaces occasionally resulted in disrupted cell sheets, only sheets harvested from PGE plates were used for scaffold seeding. Two ligamentocyte sheets were carefully wrapped around one half (length: 7.5 mm) of an embroidered P(LA-CL)/PLA scaffold functionalized through gas-phase fluorination and cross-linked collagen foam. The scaffold was placed onto the cell sheet during sheet detachment and a second sheet was transferred with a pipette onto the surface of the scaffold. Scaffolds were cultured for 7 and 14 d. 

### 2.5. Viability Assay of Cruciate Ligamentocytes in Sheets, Monolayers and Scaffolds Colonized with Sheets

Viability of ligamentocyte sheets, monolayer and scaffolds (one half of a longitudinally transected scaffold) colonized with sheets was assessed using a live/dead assay staining with fluorescein diacetate (FDA, Sigma-Aldrich) and propidium iodide (PI, Carl Roth GmbH and Ko.KG, Karlsruhe, Germany). Ligamentocytes were incubated for at least 30 s in the staining solution before starting microscopical examination. The solution consisted of 0.5% FDA and 0.1% PI dissolved in PBS. The green (living cells, cytoplasm of cells stained with FDA) or red (cell nuclei of dead cells stained with PI) fluorescence was monitored using a SPEII confocal laser scanning microscope (CLSM, Leica Microsystems GmbH, Wetzlar, Germany). The colonized area was measured based on the living cells with the ImageJ program. The cell viability was assessed based on the ratio of the area of the living cells compared to the area of the dead cells with the Image J program.

### 2.6. Histological Staining of Cell Sheets

The cell sheets were fixed for 15 min with 4% paraformaldehyde (PFA) and placed inside HistoGel (ThermoFisher Scientific Inc., Germany) before being embedded in paraffin. Sections of 7 µm were prepared. The sections were 10 min deparaffinized in xylol (Carl Roth GmbH and Ko.KG) and rehydrated with a descending ethanol row (ETOH, 99.8%, 96%, 80%, 70%) (Carl Roth GmbH and Ko.KG).

For hematoxylin-eosin (HE) staining, sections of sheets were incubated for 6 min in Harry’s hematoxylin (Carl Roth GmbH and Ko.KG), before being rinsed in running tap water and counterstained for 4 min in eosin (Carl Roth GmbH and Ko.KG).

For the alcian blue (AB) stain, sections were incubated for 3 min in 1% acetic acid. Then they were incubated for 30 min in 1% AB staining solution (Carl Roth GmbH and Ko.KG). After rinsing in 3% acetic acid and followed by a washing step in distilled water lasting 2 min, cell nuclei were counterstained with nuclear fast red aluminum sulphate solution (Carl Roth GmbH and Ko.KG) for 5 min. HE- and AB-stained sections were covered with Entellan (Merck KGaA). Images were taken using a light microscope (DM1000 LED, Leica Microsystems GmbH).

### 2.7. RNA Isolation from Ligamentocyte Monolayers, Sheets and Scaffolds Colonized with Sheets

Ligamentocyte sheets and halves of a scaffolds (7.5 mm) colonized with ligamentocyte sheets were gently rinsed in PBS before being snap-frozen and stored at −80 °C until dissected in 1-mm-sized pieces and incubated for 15 min in RLT buffer (Qiagen GmbH, Hilden, Germany) supplemented with 1% mercaptoethanol. Samples were homogenized with the tissue lyser two times for 3 minutes at 50 Hz. RNA was isolated using the RNeasy Mini kit according to the manufacturer’s recommendation (Qiagen GmbH) including an on-column DNA removal. Quantity and purity of the RNA was analyzed using the Nanodrop ND-1000 spectrophotometer (Peqlab, Biotechnologie GmbH, Erlangen, Germany) at the 260/280 absorbance ratio.

### 2.8. Quantitative Real-Time PCR

Total RNA was reversely transcribed into cDNA using the QuantiTect Reverse Transcription Kit (Qiagen GmbH) according to the manufacturer’s instructions. For each quantitative real-time PCR (qRT-PCR) the TaqMan Gene Expression Assay (Life Technologies) was used. Specific primer pairs for type I collagen (COL1A1), decorin (DCN), tenascin (TNC), Mohawk (MKX) and the reference gene glycerin-aldehyde-3-phosphate-dehydrogenase (GAPDH) were used (Table 2). qRT-PCR was performed using the real-time PCR detector StepOnePlus (Applied Biosystems (ABI), Foster City, CA, USA) thermocycler with StepOnePlus software 2.3 (ABI). Mean normalized expressions of the genes of interest in the sheets were determined in relation to the reference gene GAPDH and calculated for each specimen [39].

### 2.9. Immunofluorescence Analysis of Ligament ECM Protein Expression in Ligamentocyte Sheets and Scaffolds Colonized with Sheets

The protein expression of ligament-related components within cell sheets was detected by immunolabeling sheets and scaffolds. Immunoreactivity was detected using the CLSM. Ligamentocyte sheets or scaffolds colonized with cell sheets were fixed for 15 min in 4% PFA, before being rinsed in TRIS buffered saline (TBS: 0.05 M TRIS, 0.015 M NaCl, pH 7.6), and then, incubated with protease-free blocking solution (5% donkey serum diluted in TBS with 0.1% Triton × 100 for cell permeabilization) for 20 min at AT. Subsequently, cell sheets and scaffolds were incubated with primary antibodies suspended in blocking solution (type I collagen (1:30, goat-anti-human), decorin (1:50, rabbit-anti-human) and α-smooth muscle actin (αSMA, 1:50, mouse-anti-human) (Table 3) overnight at 4 °C in a humid chamber. Negative controls were treated in a similar manner except for omitting the primary antibodies. Sheets and scaffolds were rinsed three times with TBS before incubation with secondary antibodies (cyanine[cy]3 labeled donkey-anti-mouse or -anti-goat as well as donkey-anti-rabbit coupled with Alexa-Fluor488) diluted 1:200 in TBS overnight at 4 °C in a humid chamber. Cell nuclei were visualized using 10 µg/mL 4′,6′-diamidino-2-phenylindol (DAPI, Roche, Mannheim, Germany). In addition, some sheets were stained with phalloidin Alexa-Fluor488 (Table 3) and DAPI diluted in blocking buffer to visualize F-actin and cell nuclei. Immunolabeled or F-actin-stained cell sheets and scaffolds were rinsed three times for 5 min with TBS. Photos were immediately taken after staining using a CLSM (100 µm stack with 63 images each).

### 2.10. Statistical Analysis

Data was expressed as mean values with standard deviation. Statistics were performed using Graphpad Prism 8 (version 8.4.3, GraphPad Software Inc., San Diego, USA). Normalized and unnormalized data were evaluated using one-way ANOVA and/or one sample t-test, two-tailed. Statistical significance was set at a *p* value of ≤ 0.05. The Grubbs test was applied to identify outliers, which were excluded.

## 3. Results

### 3.1. Ligamentocyte Sheets from PGE-Coated Plates Detach More Rapidly Than Those Harvested from Uncoated Plates

Conditions for cell sheet formation such as required cell numbers depend on cell type [15]. A total of 5–8 × 10^5^ cruciate ligamentocytes per cm^2^ were at least required to allow stable cell sheet formation. After 24 h cell attachment and growth phase, the ligamentocytes at the margin of the 12-well started to detach spontaneously; thereby, a cell-free border developed (Figure 1A). The interspace between the wall of the 12-well and the cell layer was measured. This distance was significantly larger in cell culture plates with PGE coating (152.4 ± 47.7 µm) compared to the uncoated plates (99.5 ± 56.1 µm) after 24 h (Figure 1B1). Hence, there were also occasional difficulties in releasing an intact ligamentocyte sheet from the uncoated plates. The sheets harvested from the uncoated plates were more fragile, sometimes having tears or being tattered. 

Nevertheless, intact ligamentocyte sheets could be produced within 24 h with both cell culture plates with and without PGE coating, but the time needed for cell detachment from the PGE-coated plate (4.9 ± 8.1 min) compared to that required when using an uncoated plate (21.7 ± 22.7 min) was significantly lower (Figure 1B2).

### 3.2. Viability of Ligamentocytes Is Maintained in Cell Sheets

The relative viability of the sheets did not decrease significantly after detachment (day 0) and 7 d of culturing. There was no significant difference in viability in the sheets compared to the respective monolayer. In addition, it was shown that PGE coating did not change the cell viability of sheets in comparison to the uncoated 12-well plates (Figure 2A1–D).

### 3.3. Histological Examination of Cell Sheets Reveals Homogenous Cell Distribution and Sulphated Glycosaminoglycan Deposition

The ligamentocyte sheets harvested from both types of plates were investigated using HE staining to gain an overview of tissue structure. AB staining was performed to estimate sulphated glycosaminoglycan (sGAG) deposition. Observation at the timepoint immediately after detachment (day 0) revealed a high cell density with many densely packed cell nuclei and some ECM. Detached ligamentocyte sheets showed a random and dense cell arrangement with no differences between sheets from uncoated and coated surfaces (Figure 3A1–D1). However, after 7 d the cell nuclei in inner regions of the sheets were localized with a larger distance between each other and a larger amount of ECM became visible between the cells (Figure 3A1–D1 vs. Figure 3A2–D2). HE-stained cross-sections of isolated ligamentocyte sheets showed that the detached sheets were already more than 4 cell diameters thick. At 7 d the folds of the sheets had more or less fused, forming a 3D construct, still with irregular cell arrangement. This was different from the cell arrangement in rows normally detectable in the native ACL of the rabbit by HE and AB staining (Figure 3B3–D3). AB staining of the ligamentocyte sheets revealed a faint blue staining in the ECM between cells, which proved sGAG deposition already at day 0 but also later at day 7 (Figure 3C1,D1,C2,D2). sGAG deposition in sheets produced on PGE-coated and uncoated surfaces was generally weak resembling the faint staining observed in the native ACLs (Figure 3C2,D2,D3).

### 3.4. Surface Area of Sheets Decreases after Detachment and during Culturing

After detachment, the surface of the ligamentocyte sheets was significantly smaller (61.6 ± 4.3 mm^2^ with PGE coating vs. 66.1 ± 9.0 mm^2^ without coating) than the original growth area of the 12-well plates (manufacturer information for the surface of the 12 wells: 365 mm^2^ with PGE coating and 390 mm^2^ without coating) and it significantly further decreased (3.1 ± 0.8 mm^2^ with PGE coating vs. 3.6 ± 0.5 mm^2^ without coating) during the following 7-d cultivation period. At the end of the observation period the cell sheets formed a 3D construct (Figure 4A,B). F-actin stress fibers and αSMA were visualized in the monolayer and ligamentocyte sheets freshly released from the PGE-coated and uncoated plates (Figure 4C1–C3). The structure of the actin cytoskeleton was depicted to try to explain the shrinkage after sheet detachment. The expression of F-actin and αSMA did not show major differences in the sheets harvested from the PGE-coated and uncoated plates. F-actin revealed a net-like overall structure in the sheets consisting of densely packed cells (Figure 4C2,C3).

The ligamentocytes in the monolayer culture often displayed stress fibers, which were oriented mainly parallel to the longitudinal cell axis leading to cell–cell and cell-substrate contacts. αSMA was generally expressed by a subpopulation of cells. In the monolayer, it was expressed by most of the cells and was more evenly distributed over their entire cytoplasm, often co-localizing with the F-actin fibers (Figure 4C1). In contrast, αSMA was focally expressed in the ligamentocyte sheets, often only in the perinuclear rER region and only few cells showed a distribution over the entire cytoplasm (Figure 4C2,C3). 

### 3.5. Ligament-Related Genes Are Expressed in Isolated Cell Sheets Released from Surfaces with or without PGE Coating

At both observation time points (day 0, immediately after ligamentocyte sheet detachment) and day 7 the gene expression for COL1A1, DCN, TNC and MKX did not significantly differ between sheets released from PGE-coated or uncoated surfaces (Figure 5A–D). After 7 d of culturing, COL1A1 and DCN gene transcription was significantly lower in both types of cell sheets compared to the monolayer of day 0, and the COL1A1 mRNA level of the sheet derived from the uncoated plate was also significantly lower compared to the 7-day-old monolayer (Figure 5A,B). However, TNC expression was significantly higher in the monolayer (day 0) compared to both types of sheets at the same time point (day 0) (Figure 5C). Compared to the monolayer, TNC gene expression was only lower in the 7-day-old cell sheets released from the uncoated plates. Gene expression of MKX increased (day 7, not significant) in those from the PGE-coated plates compared to the monolayers of day 0 (Figure 5C,D). Nevertheless, at day 7 the expression of TNC and MKX did not significantly differ between sheets and monolayer of the same time point (Figure 5C–D).

### 3.6. Scaffolds Can Be Colonized with Ligamentocyte Sheets Harvested from PGE-Coated Surfaces

Cruciate ligamentocyte cell sheets adhered to embroidered scaffolds and cells spread onto the scaffold (Figure 6A). There was no decrease in viability of cells colonizing the scaffolds. When the viability of cells in the monolayer, cells on the scaffold and those in harvested and cultured cell sheets from PGE-coated plates were compared between 0, 7 and 14 d no significant difference could be detected (Figure 6B). The scaffold area covered with living cells increased significantly from 7 (2.7 ± 3.1%) to 14 d (11.6 ± 5.8%) (Figure 6C).

### 3.7. Ligament-Related Genes Are Expressed in Ligamentocyte Sheets and Scaffolds Seeded with Them

Relative gene expression of COL1A1 decreased significantly in cell sheets (manufactured on surfaces with PGE coating) and cultured for 7 and 14 d compared to the monolayer at 0 d (Figure 7A). However, the relative gene expression of COL1A1, DCN, TNC and MKX tended to be upregulated in scaffolds colonized by cell sheets in comparison to cell sheets alone (Figure 7A–D) and that of COL1A1, DCN and TNC was also higher in comparison to the monolayer at day 14 (Figure 7A–C).

### 3.8. Protein Expression in Ligamentocyte Sheets and Scaffolds Seeded with Them

Type I collagen and decorin protein expression could be detected in cell sheets and scaffolds seeded with ligamentocyte sheets after 7 and 14 d. It was mostly homogeneously distributed around the cells in the cell sheets with some areas of higher immunoreactivity in cell sheets of day 14 compared to day 7 (Figure 8A–C).

Type I collagen and decorin protein expression was detectable in the scaffolds seeded with ligamentocyte sheets at both time points of investigation (7 and 14 d) with more pronounced immunoreactive areas at day 14 (Figure 9).

## 4. Discussion

Thermoresponsive surfaces represent versatile tools for diverse medical applications including tissue engineering [40,41]. They allow a non-invasive enzyme-free release of the cells from the cell culture dish’s surface to gain confluent cell sheets consisting of cells and their own freshly produced ECM [42,43].

Stöbener et al. demonstrated already that cell culture devices coated with the thermoresponsive polymer PGE allowed the production of intact cell sheets from human dermal fibroblasts, human smooth muscle cells from the aorta and human umbilical vein endothelial cells [15], while the sheet detachment from standard polystyrene cell culture devices was not possible under similar conditions [14]. In the present study, lapine cruciate ligamentocyte cell sheets could be produced within 24 h, with both the PGE coating provided by Stöbener et al. [14] and in the most cases also with standard cell culture dishes. However, the time needed to release the sheets was significantly lower with the PGE-coated surfaces, and in some cases sheet production with uncoated polystyrene remained indeed unsuccessful. The required cell density of ligamentocytes needed was higher than that reported by Stöbener et al. for other cell types such as dermal fibroblasts [14]. Ligamentocytes represent the natural cell source for biofabricating ligaments. Nevertheless, in regard to ACL tissue engineering mostly mesenchymal stromal cells (MSCs) or other stem cell species were applied for sheet formation (Table 1). Hence, for ACL ligamentocytes or other tendon-derived differentiated cell types only few reference data exist [6,20]. Mitani et al. reported a seeding density of 3 × 10^4^ ACL ligamentocytes per cm^2^ but provided no information about the duration of the culture period on the thermoresponsive surface used [6]. Stem cell and progenitor cell-derived sheets can be produced based on a lower cell density (Table 1) and are characterized by abundant ECM [3,44]. This observation underlines the dependence of sheet formation on cell density required for each cell type possibly associated with ECM formation capacity, proliferation rate and contact inhibition. Hence, ascorbic acid is often added to stimulate collagen synthesis (Table 1), which stabilizes the cell sheet and allows rapid detachment without impacting the functionality of the coating. The culture medium in the present study contained 25 µg/mL ascorbic acid. The coating in the present study was very durable and could be re-used at least three times for novel sheet formation. This re-use of thermoresponsive materials (poly(NiPAAm-co-DEGMA) was already reported by Nitschke et al. with human corneal endothelial cells [45]. Directly after the 24 h cell growth phase and before detachment using PBS at AT and body temperature (37 °C) a cell-free margin could be observed at the border of the growth surface of the well near the wall, indicating the start of spontaneous detachment. This cell-free space was significantly larger in the coated compared to the uncoated dishes and a reliable sign for subsequent successful release of intact cell sheets. No intact cell sheet could be harvested if this border was lacking or gaps within the cell layer had occurred. In the present study ligamentocytes seemed to exhibit contractive forces at high cell density. The contraction ability of the cells interacting with each other and their synthesized ECM within the sheet became also evident when comparing the size of the sheets to the initial growth area. The growth area of the well was significantly larger than the released sheets. This shrinkage of the released sheets might also be a correlate of more intensive cell–cell contacts, reorganization of the cytoskeleton, e.g., contractile stress fiber formation and cell interaction with the ECM [46]. In the present study a coherent 3D F-actin network within the cell collective forming a sheet could be observed. F-actin had a rather cortical localization parallel to the interfaces of neighboring cells. Cells in the detached sheets achieved a 3D rounded cell shape resulting in smaller cell diameters compared to the monolayer, which was characterized by larger and highly flattened cells. This switch from a 2D to a 3D cell shape might have contributed to the observed shrinkage of the sheets. Nevertheless, F-actin stress fiber density did not show major differences between ligamentocyte sheets harvested from PGE-coated and uncoated plates.

Histological cross-sections of isolated ligamentocyte sheets stained directly after detachment visualized that the sheets were more than 4 cell diameters thick, hence, underlining their character of a 3D culture. To further investigate the shrinkage after sheet detachment, αSMA expression was visualized directly after sheet harvesting. αSMA represents a mechanosensitive protein, recruited in response to high tension within the stress fibers and responsible for producing high contractile forces by cells [47,48].

Obviously, in contrast to the ligamentocyte monolayers with evenly distributed αSMA, expression of the latter was decreased in the sheets. In many cells of both types of sheets it was only visible as a spot in the perinuclear rER region. This feature might be caused by the loss of cell–substrate contacts, which are probably important as anchor points for force generation, and the fact that possibly cell–cell contacts became much more important for cell communication in freshly released cell sheets. Nevertheless, this aspect has to be investigated in future in more detail. A loss of αSMA in 3D culture of ligamentocytes (in spheroids and on scaffolds) was shown previously by our research group [49]. 

After successful release, cell sheets formed spherical cell aggregates during further culturing in a non-adherent cell culture dish. Cruciate ligamentocytes are known to form aggregates when adhesion to a surface is inhibited [50]. To prevent hypoxia and subsequent cell necrosis, less than 250 µm thickness is recommended for 3D cultures [46]. Oxygen diffuses only 100–200 µm into spheroidal cell aggregates of different cell types [51]. Despite ligamentocytes derive from a bradytrophic tissue characterized by low oxygen and nutrient exchange, the hypoxia arising in the core of 3D cultures represents a risk factor for impairing cell survival [51]. Hence, necrotic zones could be expected in the core of a contracted cell sheet with a diameter of much more than 400–600 µm after 7 d of culturing [51]. They might lead to an enhanced rate of cell death and impair the number of vital cells participating in the 3D culture [50]. However, the cell viability determined was not significantly reduced.

Ligamentocyte sheets cultured for 7 days revealed a significant decrease in the gene expression of typical ECM components by the ligamentocytes such as COLA1 and DCN derived from both surfaces and in regard to TNC only from the uncoated surface in comparison to the monolayer culture at day 0. The question arises why the monolayer conditions displayed a higher gene expression for COL1A1 and DCN after 7 days when compared to the sheets (significant for the uncoated surfaces). This might be influenced by the fact that the monolayer cells were deprived of their ECM during passaging and expansion, whereas in the sheets the ECM could accumulate as shown by the histology of 7 d. The gene expression profile of typical ligament-related components, the sGAG release and cell viability did not differ between sheets from PGE-coated and uncoated surfaces. Hence the PGE surface might not significantly affect the phenotype of ligamentocytes. Unfortunately, no published reference data are available for ligamentocyte expression profile in sheets released from other thermoresponsive surfaces.

In face of the required biomechanical stability of ACL constructs produced by tissue engineering resilient biomaterials such as PLA or silk fibers, suitable processing techniques are required [52,53]. Cell sheet-derived biomaterial-free constructs might be too weak at all [33,54]. Nevertheless, tendon and ligament tissue engineering utilizing cell sheets is an emerging research field [6,33,54]. The resistance to tension is substantially lower in constructs produced by cell sheets compared to the native tissue [2,55]. The use of biomechanically suitable scaffolds can overcome this limitation. Embroidered scaffolds manufactured from bio- and cyto-compatible polymers such as P(LA-CL) and PLA could fulfill these requirements, since the embroidery pattern can be adapted to biomechanical demands [50,56]. Whether such a scaffold combined with a cell sheet has other biomechanical properties than a scaffold alone remains unclear and should be addressed in future studies. To improve cell adhesion to P(LA-CL)/PLA scaffold surfaces gas fluorination and a native collagen foam cross-linked with HMDI could be used for functionalization, which showed promising results in previous studies [38,57]. 

The ligamentocyte sheets adhered well to the embroidered P(LA-CL)/PLA scaffold and the cells spread subsequently, whereby the surface of the scaffold covered by cells increased significantly with time, comparing 7 and 14 d. Due to the shrinkage of the sheets placed on the scaffold compared to the detached sheet, after 7 d less than 5% of the surface was covered by cells. However, one week later after 14 d this area was more than duplicated, being more than 10% mediated by cell spreading and migration onto the fibers of the scaffolds. Ligamentocyte emigration from 3D cultures on a scaffold consisting of synthetic fibers has been described previously [38,49]. The underlying cell signaling processes initiating cell emigration and the possible additional contribution of an enhanced cell proliferation at the interface between sheets and scaffold have to be investigated in future in more detail. Since the viability did not change, longer culturing periods are advised. The ligamentocytes still expressed the main ligament ECM components COL1A1 and DCN on the scaffolds as well as the fibroblast marker TNC and the transcription factor MKX on the gene level. Type I collagen is of central importance for its resiliency in regard to tension [58]. The elevated COL1A1 gene expression, observed on the P(LA-CL)/PLA scaffolds after 14 compared to 7 (scaffold) and 0 d (monolayer) might show the onset of tissue formation [59]. The protein expression analysis suggests a more pronounced immunoreactivity for type I collagen and decorin at 14 compared to 7 d, which is in accordance with a previous study using the same cell type and similar scaffolds but a cell suspension for seeding and dynamical culture [38]. Decorin is the most important proteoglycan in tendons and ligaments with a life-long expression and a contribution to tendon healing [60,61]. It also regulates the collagen fibrillogenesis, the modulation of growth factor activity and cell growth [62], and it was designated as indicator for tendon development [63,64]. The DCN gene expression in P(LA-CL)/PLA scaffolds seeded with ligamentocyte sheets showed after 14 d a substantial increase compared to the monolayer at day 0. In addition, the trend of an increase in decorin protein and gene expression after 14 d of culture compared to 7 d became evident. However, the dynamical culture of the same scaffolds seeded with suspended lapine ACL ligamentocytes did not show any increase in DCN gene expression between 7 and 14 d of culturing [38].

The gene expression of the fibroblast marker TNC, which represents an extracellular glycoprotein, also involved in the maintenance of fibrocartilaginous phenotypes by reducing cell–ECM adhesion [65] and contributing to tissue elasticity [66], was amplified in the scaffolds after 14 compared to 7 (scaffold) and 0 d (monolayer). The dynamical culture of the same scaffolds with suspended lapine ACL ligamentocytes showed rather a decrease of TNC expression after 14 compared to 7 d [38].

The ligament-related transcription factor Mohawk, known to be involved in regulation of COL1A1 gene expression [67], increased in the scaffolds between 7 and 14 d. The dynamical culture of the same scaffolds with suspended ligamentocytes showed the same trend in the MKX gene expression [38]. This suggests, together with the increasing expressions of COL1A1, DCN, TNC, MKX from 7 to 14 d, the maintenance of the ligament-related phenotype in the scaffolds. Taken together, the stable expression profile of ligament-related components in the scaffolds seeded with sheets can be highlighted here. The advantage of the novel PGE coating compared to commercially available thermoresponsive surfaces is that it does not affect cell adhesion and phenotype. An interference of the commercially available PNIPAm coating with mouse fibroblast adhesion was reported previously by Becherer et al., [27] and for smooth muscle fibroblasts by Stöbener et al. [15]. Compared to PNIPAm-coated dishes with a detachment time of around 30 min [6] the detachment time on a PGE-coated surface is substantially shorter (5 min). The preparation of PGE coatings is very easy and can be done by the researcher on his or her own without sophisticated equipment and at low costs. The devices required include only pipettes, water, ethanol and a simple ultraviolet light lamp.

However, the handling of fragile cell sheets is generally challenging [68,69,70]. Special devices for a safer transfer of sheets have been recommended [68,71,72,73,74] including in situ gelation techniques using gelatin hydrogel carrier membranes [69], commercially available Immobilon-P^®^ and SUPRATHEL^®^ membranes [70] or electrospun fiber mats made of polycaprolactone functionalized with thrombin [75]. Envelopment of scaffolds with cell sheets is indeed time-consuming as experienced in the present study and could be optimized in future using, e.g., hydrogels to achieve an improved surface colonization of the scaffolds. Nevertheless, one future aim is to generate zonal ACL tissues using cell sheet technology. Reconstruction of the ACL requires a stable attachment into the bone [76,77,78].

Fibrocartilaginous transition and bony zones are necessary at both ends of the tissue-engineered ACL construct to mediate firm ligament-to-bone attachment [77,79]. Cell sheets could present a strategy of directed seeding of these three cell types (ligamentocytes, fibrochondrocytes and osteoblasts) found within the enthesis on the same scaffold.

## 5. Conclusions

In this study we established pure cruciate ligamentocyte sheets that maintained ligament-related marker expression. Coating of cell culture plates with PGE allowed a very rapid sheet production within 24 h and the sheet detachment required in mean only 5 min. Primary cells of different cell donors allowed comparable sheet formation underlining the reproducibility. Viability and gene expression profile of ligamentocytes in sheets harvested from PGE-coated plates did not significantly differ from cell performance in sheets harvested from uncoated plates. Moreover, these sheets represent a novel strategy for directed seeding of embroidered scaffolds. The expression of ligament-related components was enhanced on the scaffolds seeded with cell sheets compared to the cell sheets alone and monolayers consisting of cells from the same donor. In this case P(LA-CL)/PLA scaffolds were proven here as suitable to be seeded with cell sheets. Thus, the cell sheet-seeded P(LA-CL)/PLA scaffolds can be used for ACL tissue engineering, improving the biological outcomes of using cell sheets alone.

## Figures and Tables

**Figure 1 cells-10-00877-f001:**
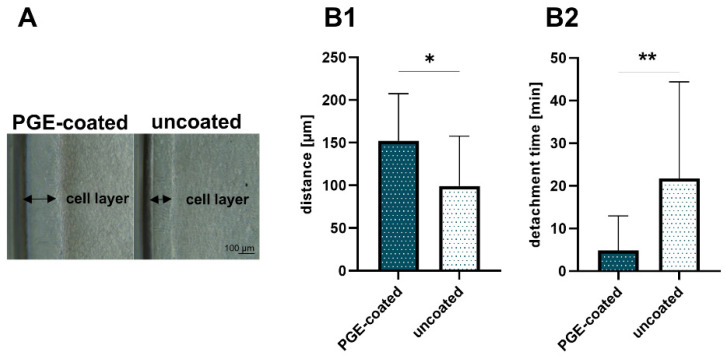
Self-detachment before thermal treatment and detachment time after treatment. (**A**) Cell-free border in poly(glycidyl ether) (PGE)-coated and uncoated control plates. (**B1**) Distance between well wall and margin of the ligamentocyte layer measured in PGE-coated and uncoated plates, n = 3 independent experiments with ligamentocytes of different donors. (**B2**) Detachment time in PGE-coated and uncoated plates, n = 4 independent experiments with ligamentocytes of four different donors. Scale bar: 100 µm (**A**). Statistic: One sample t-test, two-tailed (*). *p* values: * <0.05, ** <0.01 (**B1**,**B2**).

**Figure 2 cells-10-00877-f002:**
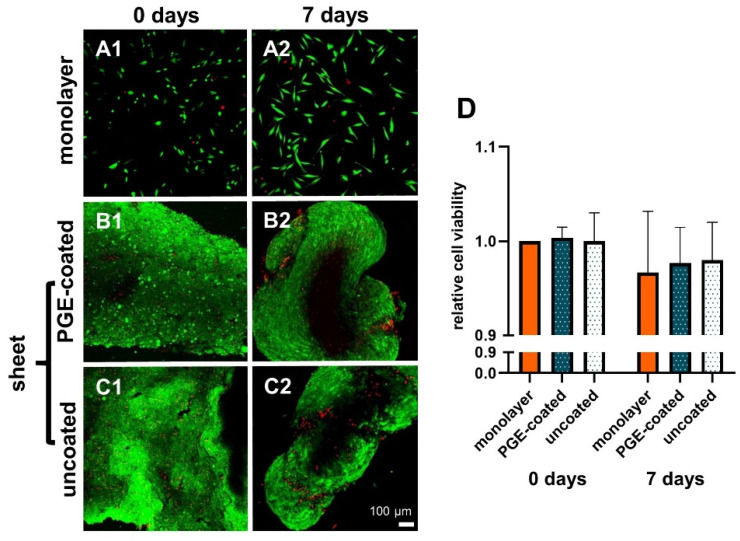
Viability of ligamentocytes in monolayer used for sheet preparation and respective sheets prepared on PGE-coated and uncoated plates and cultured for 7 d in non-adherent culture dishes. 0 d (**A1**,**B1**,**C1**), 7 d (**A2**,**B2**,**C2**), monolayer of the same cell donor (**A1**,**A2**). Poly(glycidyl ether) (PGE)-coated plates (**B1**,**B2**), uncoated plates (**C1**,**C2**). Green: vital, red: dead cells (**A1**–**C2**). (**D**) Measurements of relative cell viability in monolayer, PGE-coated and uncoated plates after 0 and 7 d, n = 3 independent experiments with ligamentocytes of three different donors. Scale bar: 100 µm (**A1**–**C2**). Statistic: One sample t-test, two-tailed (comparison with day 0 monolayer), paired one-way ANOVA (comparison between the groups) (**D**).

**Figure 3 cells-10-00877-f003:**
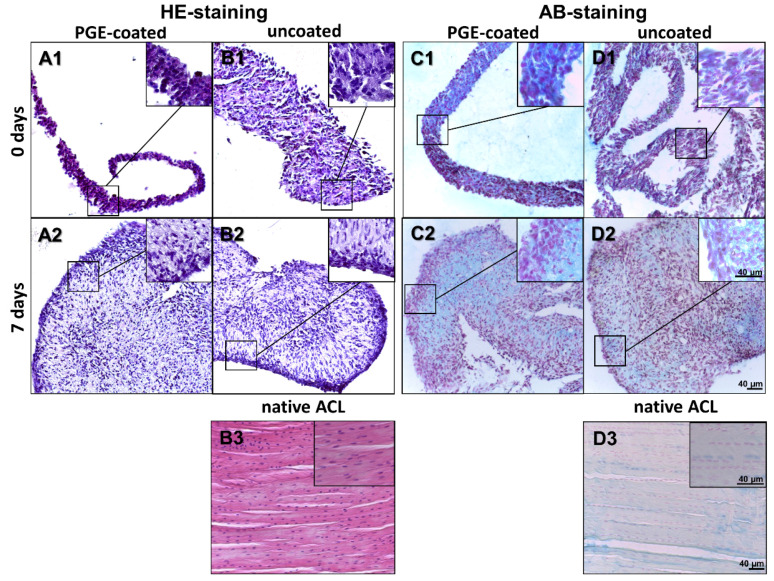
Histology of lapine ligamentocyte sheets (0 d) released from PGE-coated and uncoated plates and after culturing for 7 days. Paraffin sections of sheets released from surfaces coated with poly(glycidyl ether) (PGE) or uncoated plates stained with hematoxylin-eosin (HE, **A1, B1,A2,B2**) or alcian blue (AB, **C1,D1,C2,D2**). 0 d (**A1**,**B1**,**C1**,**D1**), 7 d (**A2**,**B2**,**C2**,**D2**). Native lapine anterior cruciate ligament (HE: **B3**, AB: **D3**), scale bars: 40 µm.

**Figure 4 cells-10-00877-f004:**
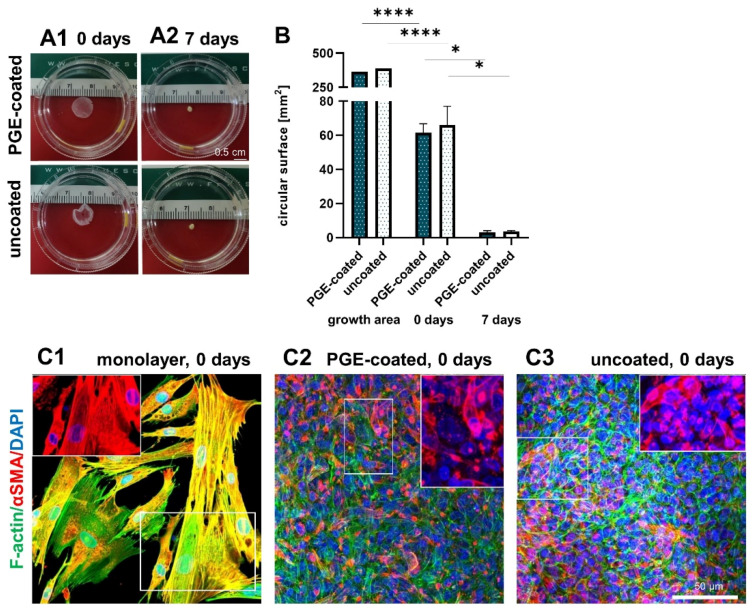
Shape and size of the sheets after detachment (0 d) and when cultured for 7 d. Shape of ligamentocyte sheets after 0 d (**A1**) and 7 d (**A2**) transferred to a Petri dish. (**B**) Circular surface of sheets released from poly(glycidyl ether) (PGE)-coated and uncoated plates after 0 and 7 d in comparison to the growth area of one well of the respective 12-well plate. n = 3 independent experiments with ligamentocytes of three different donors. (**C**) F-actin (green) and αSMA (red) of the monolayer (**C1**) and sheets released from PGE-coated (**C2**) and uncoated (**C3**) plates. The insets show exclusively αSMA at higher magnification (red). Cell nuclei were counterstained using 4’,6-diamidino-2-phenylindole (DAPI, blue). Scale bars: 0.5 cm (**A1**–**A3**), 50 µm (**C**). (**B**) Statistic: unpaired one-way ANOVA (comparison between the groups) (*). *p* values: * <0.05, **** <0.0001.

**Figure 5 cells-10-00877-f005:**
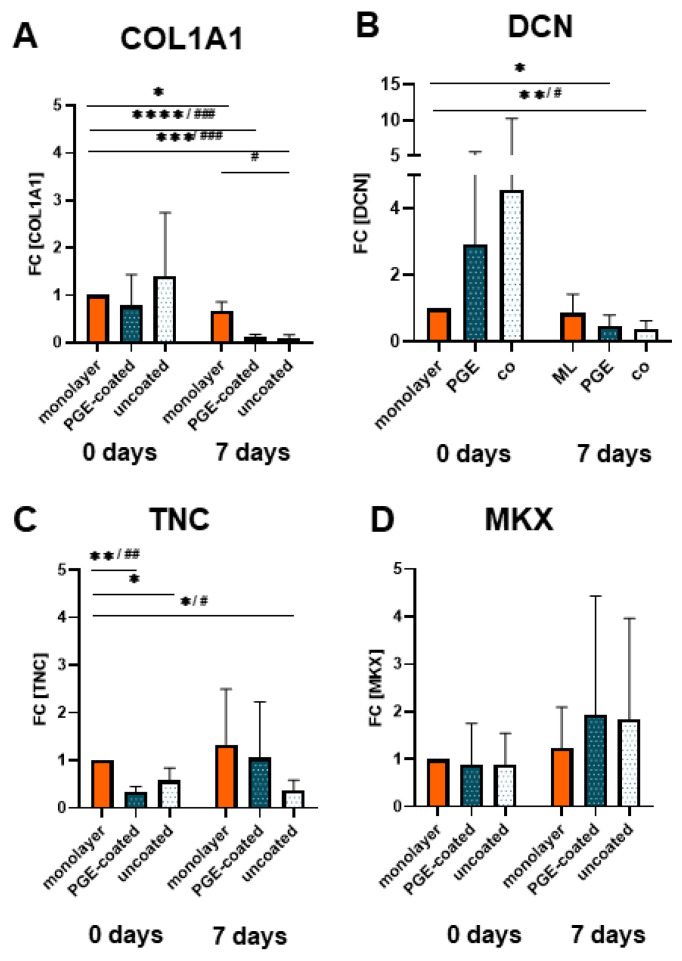
Gene expression of ligamentocytes cultured in the monolayer or as cell sheets after detachment (0 d) and when cultured for 7 d. (**A**) Type I collagen (COLA1, type I collagen A1 chain), (**B**) decorin (DCN), (**C**) Tenascin C (TNC) and (**D**) Mohawk (MKX) expression in monolayer, poly(glycidyl ether) (PGE)-coated and uncoated plates after 0 and 7 d, n = 4 independent experiments with ligamentocytes of four different donors. Statistic: One sample t-test, two-tailed (comparison with day 0 monolayer) (*), paired one-way ANOVA (comparison between the groups) (#), Grubbs test (α = 0.05). *p* values: */# <0.05, **/## <0.01, ***/### <0.001, ****/#### <0.0001 (**A**–**D**).

**Figure 6 cells-10-00877-f006:**
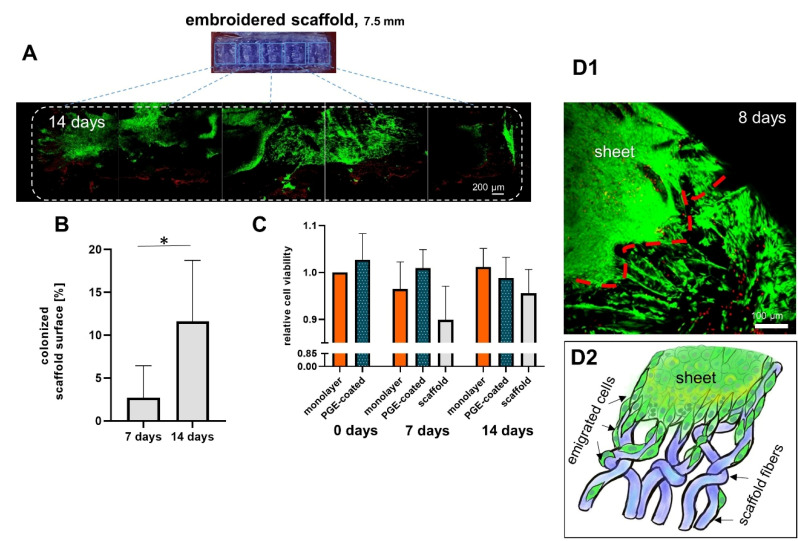
Ligamentocyte viability and colonized scaffold surface area of ligamentocytes emigrated from ligamentocyte sheets on a P(LA-CL)/PLA scaffold prepared with PGE-coated dishes after 7 and 14 d. (**A**) Colonization of the scaffold by ligamentocytes is shown (14 d, a half of the scaffold is shown). Green: vital, red: dead cells. **B**) Colonized scaffold surface after 7 and 14 d. Comparison of cell viability in monolayer, isolated cell sheets released from poly(glycidyl ether) (PGE)-coated plates and colonized scaffolds, n = 3 independent experiments with ligamentocytes of three different donors. (**C**) Relative ligamentocyte viability, n = 4 independent experiments with ligamentocytes of four different donors. ((**D1**) Cells emigrating from a cell sheet onto P(LA-CL)/PLA scaffolds after 8 days at higher magnification (red line = border of the sheet) and (**D2**) a scheme of cell emigration from a sheet on scaffold fibers. (**A**) Scale bar: 200 µm. Statistic: (**B**) One sample t-test, two-tailed (comparison with day 0 monolayer), paired one-way ANOVA (comparison between the groups). (**C**) One sample t-test, two-tailed (*), *p* values: * <0.05.

**Figure 7 cells-10-00877-f007:**
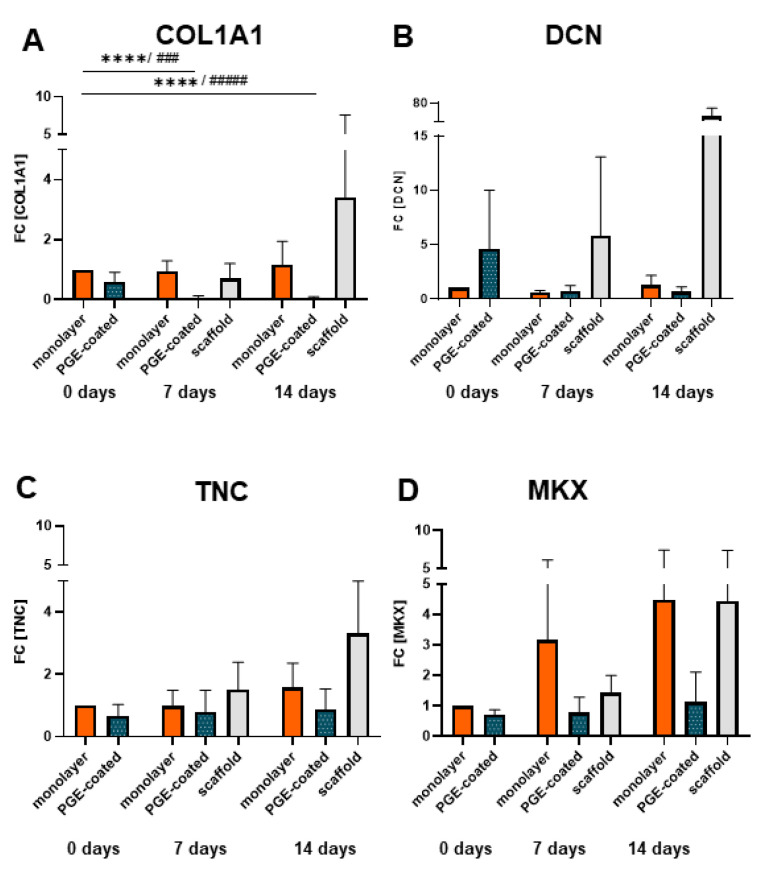
Gene expression of ligamentocytes cultured in monolayers, in sheets harvested from PGE-coated plates or in the scaffolds seeded using sheets after detachment (0 d), 7 and 14 d. (**A**) Type I collagen (COLA1), (**B**) decorin (DCN), (**C**) Tenascin C (TNC) and (**D**) Mohawk (MKX) gene expression in monolayer, poly(glycidyl ether) (PGE)-coated and uncoated plates after 0, 7 and 14 d, n = 4 independent experiments with ligamentocytes of four different donors. Statistic: One sample t-test, two-tailed (comparison with day 0 monolayer) (*), paired one-way ANOVA (comparison between the groups) (#), Grubbs test (α = 0.05). *p* values: ### <0.001, ****/#### <0.0001 (**A**–**D**).

**Figure 8 cells-10-00877-f008:**
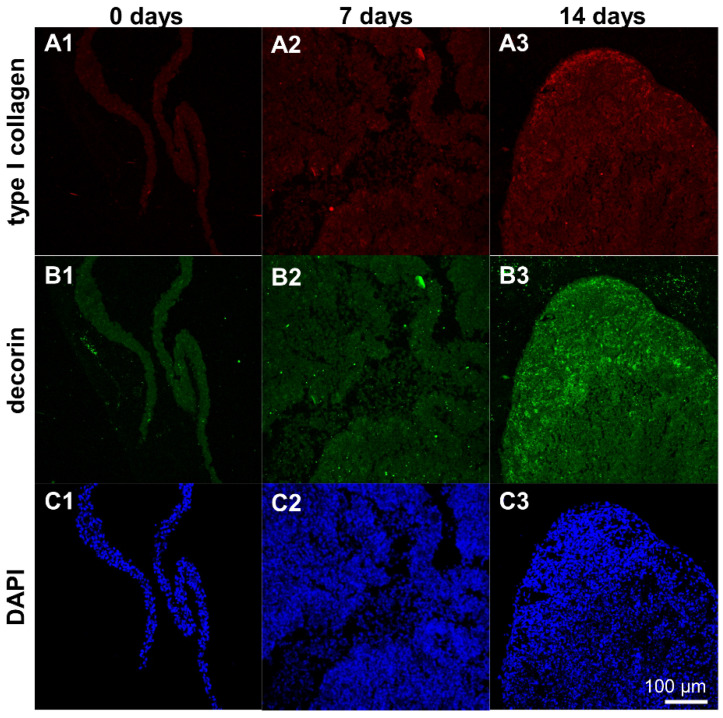
Protein expression of type I collagen and decorin by ligamentocyte sheets released from PGE-coated surfaces after detachment (0 d) and when cultured for 7 or 14 d. Type I collagen is depicted in red (**A1**–**A3**), decorin in green (**B1**–**B3**). Ligamentocyte nuclei were counterstained in blue using 4’,6-diamidino-2-phenylindole (DAPI) (**C1**–**C3**). A view on the sheet is shown by confocal laser scanning microscopy. Scale bar: 100 µm (**A1**–**C3**).

**Figure 9 cells-10-00877-f009:**
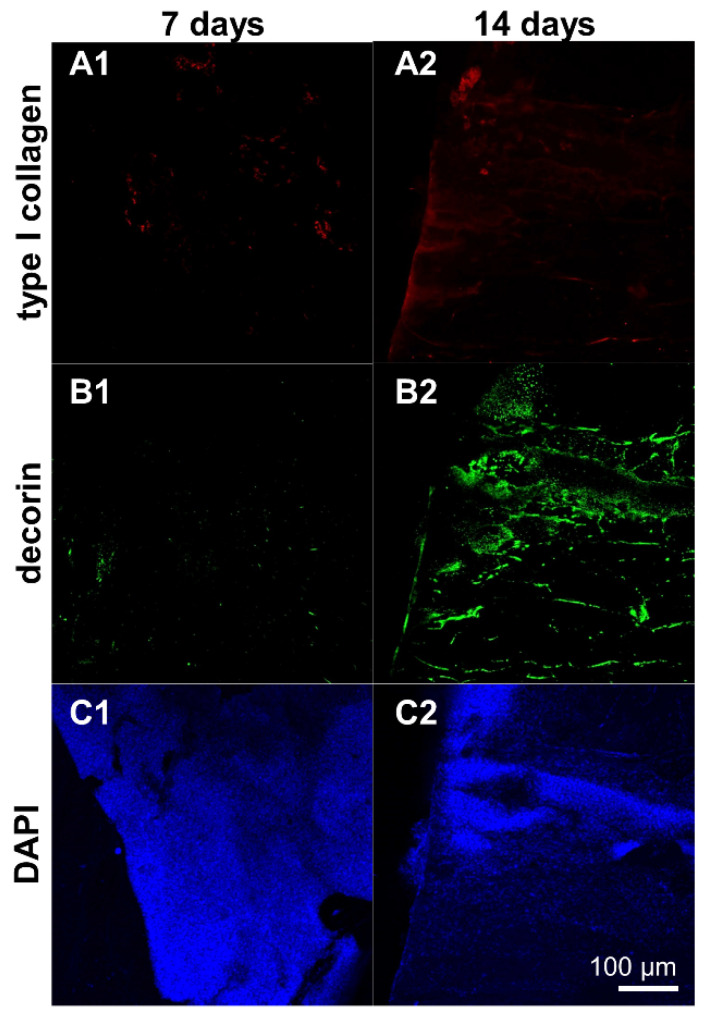
Protein expression of type I collagen and decorin by ligamentocyte sheets released from PGE-coated surfaces and seeded on embroidered scaffolds for 7 or 14 d. Type I collagen is depicted in red (**A1**,**A2**), decorin in green (**B1**,**B2**). Ligamentocyte nuclei were counterstained in blue using 4’,6-diamidino-2-phenylindole (DAPI) (**C1**,**C2**). A view on the sheet is shown by confocal laser scanning microscopy. Scale bar: 100 µm (**A1**–**C2**).

**Table 2 cells-10-00877-t002:** Primers used to assess gene expression.

Gene Symbol	Species	Gene Name	NCBI Gene Reference	Efficacy	Amplicon Length (bp)	Assay ID *
COL1A1	*O. cuniculus*	type I collagen	AY633663.1	1.94	70	Oc03396073_g1
DCN	*Homo sapiens*	decorin	NM_133503.3	2.03	77	Hs00370384_m1
GAPDH	*O. cuniculus*	glycerin-aldehyde-3-phosphate-dehydro-genase	NM_001082253.1	1.95	82	Oc03823402_g1
MKX	*O. cuniculus*	Mohawk	XM_002717295.1	1.83	60	Oc06754037_m1
TNC	*O. cuniculus*	tenascin C	FJ480400.1	1.83	61	Oc06726696_m1

*O.: Oryctolagus* * All primers from Applied Biosystems^®^ (life technologies ^TM^).

**Table 3 cells-10-00877-t003:** Antibodies and staining used to assess protein expression in cell sheets.

Target	Primary Antibody	Dilution	Secondary Antibody	Dilution
collagen type I	goat-anti-human (COL1A1 chain), Abcam, Cambridge, UK	1:50	donkey-anti-goat; cy 3, Dianova GmbH, Hamburg, Germany	1:200
decorin	rabbit-anti-human, OriGene Rockville, MD, USA	1:50	donkey-anti-rabbit; Alexa-Fluor488, ThermoFisher Scientific Inc., Germany	1:200
phalloidin Alexa- Fluor488	stains filamentous (F-) actin, Santa Cruz Biotechnology, Inc, Dallas, TX, USA	1:200	-	-
α-smooth muscle actin	mouse-anti-human, Sigma-Aldrich (A5228), Munich, Germany	1:50	donkey-anti-mouse; cy3, Dianova GmbH, Hamburg, Germany	1:200

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
