# Peer review of "Cruciate Ligament Cell Sheets Can Be Rapidly Produced on Thermoresponsive poly(glycidyl ether) Coating and Successfully Used for Colonization of Embroidered Scaffolds"

_cells, 2021, doi:10.3390/cells10040877_

Round 1
Reviewer 1 Report
This manuscript described a method for creating cell sheets of ligament fibroblasts and applying them to lpolymeric ligament scaffolds. This is presented as a useful option for populating tissue engineered scaffolds.
The work is interesting, but the methods and some of the results are vague. Differences between the number and quality of cell sheets obtained with and without the use of the biomaterial are lacking.
The temperature used for detachment and the time of application could be stated more clearly in the manuscript.
There is no explicit data describing how the cells spread after being applied to the scaffold other than one sentence in the discussion section, there should be explicit measurements because this is important for tissue regeneration.
Does figure 9 display a cross-section of the scaffold or a view along the length?
There should be more information describing the difficulty of manufacturing cell sheets without PGE.
A picture of the scaffold would be helpful. A schematic showing how the cell sheets were attached would help.
How many cell sheets were applied to the scaffold, was it just one? More detail about this process would be appreciated.
More information on the biomaterial free cell sheets in general is necessary.
Author Response
Dear Editor, 27th March 2021
The authors would like to thank the reviewer again for carefully reading the manuscript and very valuable comments. We modified the manuscript according to the reviewer suggestions with a list of changes shown below. We added some novel data and citation to address the reviewer comments. All changes performed are indicated in red and underlined in the revised version of the manuscript. We hope you will find this manuscript suitable for publication in “Cells”. Please do not hesitate to contact me anytime for questions regarding this manuscript.
Sincerely,
Univ.-Prof. Dr. Gundula Schulze-Tanzil
(corresponding author)
Reviewer 1
This manuscript described a method for creating cell sheets of ligament fibroblasts and applying them to lpolymeric ligament scaffolds. This is presented as a useful option for populating tissue engineered scaffolds.
The work is interesting, but the methods and some of the results are vague.
Response: We described the methods in more detail adding methodological details (e.g. 2.4 and 2.9) and improved also the result section (e.g. 3.1., 3.3., 3.4.).
Differences between the number and quality of cell sheets obtained with and without the use of the biomaterial are lacking.
Response: We describe the difference in the manuscript now: Sometimes the sheets harvested from the uncoated plates had some tears or were tattered (see 3.1., lines 353-355). The problems which were observed sometimes with the control plate and which methodical variation was used to manage them are also stated in 2.4 (lines 223-229).
The temperature used for detachment and the time of application could be stated more clearly in the manuscript.
Response: It is explained in more detail now (see section 2.4. lines 221-222).
|
|
Figure 1: Depicted is an example of cell sheets, in this case, produced from lapine chondrocytes either using a PGE-coated plate (left) or a uncoated plate at the same observation time of detachment. It shows that the sheet on the PGE coated plate had completely detached whereas the sheet on the uncoated plate is incompletely released and requires further handling for successful release. With ligamentocytes, we observed the same feature but did, unfortunately, not take images. |
There is no explicit data describing how the cells spread after being applied to the scaffold other than one sentence in the discussion section, there should be explicit measurements because this is important for tissue regeneration.
Response: We expressed it as a measurement in figure 6B where the colonized scaffold surfaces more than doubled their sizes between 7 and 14 days. The observation that ligamentocytes emigrate from 3D cultures such as spheroids placed on a scaffold and subsequently colonize a scaffold has already been published by our group (Hahner et al., 2015 Journal of Biomaterials Science, Polymer Edition; Gögele et al., 2020_reference 38, Schwarz et al., 2019_reference 49). Hahner et al., (2015) measured the migration distance from spheroids. We performed previously experiments with sheets with other musculoskeletal cell types and various scaffolds (Polyglycolic acid, Polylactic acid, unpublished results) which nicely depict how cells emigrate as shown below. The cells attach to the fibers of the scaffold and migrate from the sheet on the fibers thereby leaving the sheet. We added this aspect in the discussion section (lines 526-531 and inserted an image (Figure 6D) depicting ACL ligamentocytes emigrating from a sheet onto P(LA-CL)/PLA scaffold together with an explaining scheme.
|
|
|
Figure 2: Sheets performed with PGE-coated plates and seeded onto a polyglycolic acid (PGA) scaffold for 7 (left, using lapine chondrocytes) and 10 days (middle, using human mesenchymal stromal cells). Viable cells are green and dead cells as well as PGA fibers are depicted red. Cells emigrating from the sheets onto the scaffold fibers are clearly visible. On the right are lapine anterior cruciate ligamentocytes shown emigrating after 20 d from a cell sheet placed on a poly L lacitic acid scaffold. |
Does figure 9 display a cross-section of the scaffold or a view along the length?
Response: It is stated now in the legend of Figs. 8 and 9 that a view on the sheets or scaffolds with sheets is shown.
There should be more information describing the difficulty of manufacturing cell sheets without PGE.
Response: Cell numbers and density required for sheet formation was the same. The difficulty were the efforts required in successful harvesting of intact sheets and their stability. It is described now in section 2.4. concerning the release of intact sheets. We refer also to the statement in the discussion section (lines 455-456) (see fig. 1 of this point by point reply).
A picture of the scaffold would be helpful. A schematic showing how the cell sheets were attached would help.
Response: We established a schematic showing how the cell sheets attached. We added it to the graphical abstract. We improved the schematic drawing of the graphical abstract.
How many cell sheets were applied to the scaffold, was it just one? More detail about this process would be appreciated.
Response: We applied two sheet to the scaffold. This important information and the procedure of seeding sheets onto the scaffold is mentioned now in 2.4.
More information on the biomaterial free cell sheets in general is necessary.
Response: I guess the reviewer means the sheets derived from the uncoated plates not used for scaffold seeding. The differences in harvesting and handling are stated in 3.1. of the result section. We added F-actin and α-smooth muscle actin (SMA) staining now to compare the cytoskeletal architecture in both types of sheets (from PGE-coated and uncoated plates) immediately after detachment more thoroughly. We compared it also with the architecture in the monolayer culture (novel Fig. 4C).

Reviewer 2 Report
Reviewer comments:
This paper describes a study in which ligamentocytes cell sheets are produced in PGE-coated surfaces and compared to the same uncoated. Moreover, the potential of functionalized PLA/(P(LA-CL)) scaffolds seeded with the cell sheets was also screened. This work is of scientific relevance, since a new type of cell sheets based on ligamentocytes is proposed and analyzed when combined with scaffolds for ACL tissue engineering. However, the manuscript lacks of adequate English language and there are major points that should be revised/addressed as I describe below.
Abstract
- “ Ligamentocytes were seeded for 24 h on surfaces either coated with or without PGE”. Authors should revise the sentence. It is not clear the type of surfaces that authors mention. Also, “coated with or without PGE”. What do you mean by coating without PGE? I believe that authors intend to say “Ligamentocytes were seeded for 24 h on surfaces coated with PGE or without coating.”
- Biological characterization was performed for “sheets (with/without PGE)”. Do you mean for sheets detached from the “surfaces” coated with PEG and without coating, and that were further were cultured separately? Also, what do you mean by “scaffolds (with PGE) and the respective monolayers”? From the previous paragraph authors said that ligamentocyte sheets were cultured around an embroidered scaffold of PLA and P(LA-CL), but it is no clear that these scaffolds were coated with PEG.
- Authors should revise the abstract to make it more clear as a first reading of the manuscript without any knowledge of the study.
Introduction
- Introduction title is missing.
- Authors mentioned the commercial PNIPAm-coated dishes used to harvest sheets of different cell types including for ligament sheet formation. If a commercial product has been used with success for ligament sheet formation, what is the major advantage of the system proposed in the article as compared to the commercial. If the commercial has flaws this should be mentioned, in a way that the PGE-based system may overcome them.
Material and Methods
- 2.6. Authors should revise the term “Vitality” to “Viability”, as well as, throughout the manuscript text.
- Authors should include gene expression sections (2.8, 2.9) first than the immunofluorescence analysis (2.7), as the gene expression profile comes first than the protein expression (as authors described in results section).
Results
- 3.1. In figure 1, the control “co” should be identified as “uncoated” and the “PGE” as “PGE-coated”.
- 3.2. “Observation at the timepoint of detachment (day 0), which means after 24 h culturing on the plate and immediately after detachment revealed a high cell density with many densely packed cell nuclei and some ECM. Sheets showed a random and dense cell arrangement with no differences between sheets from uncoated and coated surfaces (Fig. 2A1-D2).” I believe that the images described are Fig. 2A1-D1.
- 3.2. Please correct: “AB staining revealed a faint blue staining in the ECM between cells which proved sGAG deposition already at day 0 but more pronounced at day 7 (Fig. 2C1-D2).”
- 3.3. As in the materials and methods section, authors should replace “vitality” by “viability”
- 3.3. In figure 3, authors should also replace “co” by “uncoated” and “PGE” by “PGE-coated”. As for “ML”, “Monolayer” would be preferable.
- It would make more sense if the information regarding from cell viability (3.3.) come first than that from cell distribution and ECM formation obtained from HE staining and AB (3.2). This is also applicable for material and methods section.
- 3.4. In figure 4, authors should also replace “co” by “uncoated” and “PGE” by “PGE-coated”.
- 3.4. Authors should provide a better explanation about the “original/initial growth area of the 12-well plate”. Apparently, the wells were also coated and uncoated with PGE. Explain this either. I do not have access to graphical abstract.
- 3.4. In figure 4A authors should also include images of the remaining conditions, uncoated and original growth area. The same rational for figure 4C, and authors should include the F-actin staining protocol in the material and methods.
- 3.5. In figure 5, authors should also replace “co” by “uncoated”, “PGE” by “PGE-coated”, and “ML” by “Monolayer”.
- 3.5. “At both observation time points (day 0, immediately after sheet detachment) and day 7 the gene expression for COL1A1, DCN, TNC and MKX did not significantly differ between sheets released from PGE-coated or uncoated surfaces and monolayer (Fig. 5A-D).” This statement is not consistent to the observations from figure 5, and also to the text that followed.
- 3.5. Please rephrase: “TNC gene expression was only lower in the 7-day-old cell sheets released from the uncoated plates and that of MKX was lower in those from the PGE-coated plates, both compared to the monolayers of day 0 (Fig. 5C, D).” The sentence is quite confusing.
- 3.6. “There was no decrease in viability of cells colonizing the scaffolds. When the viability of cells in the monolayer, cells on the scaffold and those in harvested and cultured cell sheets from PGE-coated plates were compared between 0, 7 and 14 d no significant difference could be detected (Fig. 6B).” Cell viability was detected by live/dead staining, and for that reason authors should consider to standardize the term to “viability” and not “vitality” as described in other sections.
- 3.6. Why did authors used the monolayer cultures as control for comparing with PGE-coated and the scaffolds and not also the uncoated condition as in the previous sections?
- 3.6. In figure 6C, authors should also replace “PGE” by “PGE-coated”, “ML” by “Monolayer”, and “S” by “Scaffold”
- 3.7. Once again the uncoated condition was excluded. My question goes also for the fact that in previous sections was observed that no significant improvements were observed after coating the surfaces with PGE. In fact, globally the uncoated condition showed better results than that observed for the PGE-coated obtained cell sheets.
- 3.7. In figure 7, authors should use more thick lines to indicate the statistical differences. The same for figure 5.
Discussion
- I agree that the PGE-coated surfaces induce a more stable detachment of cell sheets than the uncoated surfaces, and authors highlighted this as a major advantage of the system. However, from histological analysis it was observed and stated by the authors that the cell sheets recovered from uncoated surfaces presented a higher ability to produce sulphated GAGs typical of these type of cells producing ECM. Authors could comment on that.
- “The use of biomechanically suitable scaffolds can overcome this limitation.” It would be interesting if authors could provide some tensile characterization tests of scaffolds, before and after the presence of cell sheets.
- From gene expression profile of figure 7, the monolayer condition presented for some markers significantly higher expression than the PGE-coated and uncoated conditions, as also described by the authors in results section. This should also be discussed by the authors.
- Authors should better evidence the fact that the scaffolds were able to improve the expression of important markers related to ACL formation and function. No so much from protein expression, but in terms of gene expression this was quite evident, and should be better emphasized by the authors as an improved strategy for ACL tissue engineering, complementing and improving the outcomes of using cell sheets alone. Maybe the biomechanics of the scaffolds is influencing the results, and some results on that would be interesting.
Conclusions
- This section should be significantly improved. Authors basically state that there was no advantages of coating the surfaces with PGE, which was one of the main purposed of the study. Also, the improvements achieved with scaffolds should be mentioned or better highlighted. Authors, should mention the type of scaffolds used, and at least refer the PGE-coating vs. uncoating main achievements.
Author Response
Dear Editor, 27th March 2021
The authors would like to thank the reviewer again for carefully reading the manuscript and very valuable comments. We modified the manuscript according to the reviewer suggestions with a list of changes shown below. We added some novel data and citation to address the reviewer comments. All changes performed are indicated in red and underlined in the revised version of the manuscript. We hope you will find this manuscript suitable for publication in “Cells”. Please do not hesitate to contact me anytime for questions regarding this manuscript.
Sincerely,
Univ.-Prof. Dr. Gundula Schulze-Tanzil
(corresponding author)
Reviewer 2
This paper describes a study in which ligamentocytes cell sheets are produced in PGE-coated surfaces and compared to the same uncoated. Moreover, the potential of functionalized PLA/(P(LA-CL)) scaffolds seeded with the cell sheets was also screened. This work is of scientific relevance, since a new type of cell sheets based on ligamentocytes is proposed and analyzed when combined with scaffolds for ACL tissue engineering.
However, the manuscript lacks of adequate English language and there are major points that should be revised/addressed as I describe below.
Response: We improved the language and adressed all issues raised by the reviewer listed below point by point.
Abstract
- “Ligamentocytes were seeded for 24 h on surfaces either coated with or without PGE”. Authors should revise the sentence. It is not clear the type of surfaces that authors mention. Also, “coated with or without PGE”. What do you mean by coating without PGE? I believe that authors intend to say “Ligamentocytes were seeded for 24 h on surfaces coated with PGE or without coating.”
Response: We thank the reviewer for the proposed much clearer formulation and adopted it.
- Biological characterization was performed for “sheets (with/without PGE)”. Do you mean for sheets detached from the “surfaces” coated with PEG and without coating, and that were further were cultured separately? Also, what do you mean by “scaffolds (with PGE) and the respective monolayers”? From the previous paragraph authors said that ligamentocyte sheets were cultured around an embroidered scaffold of PLA and P(LA-CL), but it is no clear that these scaffolds were coated with PEG.
Response: We apologize for our misleading formulation and thank the reviewer for indicating this. Lines 43-44: we write now: „detached from surfaces with/without PGE coating“ and „scaffolds seeded with sheets from PGE-coated plates“.
- Authors should revise the abstract to make it more clear as a first reading of the manuscript without any knowledge of the study.
Response: We revised the abstract according to the reviewer comments.
Introduction
- Introduction title is missing.
Response: We inserted two subheadings in the introduction section („Cell sheet technology“ in the beginning and „Thermoresponsive coatings for cell sheet formation“)
- Authors mentioned the commercial PNIPAm-coated dishes used to harvest sheets of different cell types including for ligament sheet formation. If a commercial product has been used with success for ligament sheet formation, what is the major advantage of the system proposed in the article as compared to the commercial. If the commercial has flaws this should be mentioned, in a way that the PGE-based system may overcome them.
Response: The novel coating is versatile to be used also for other applications such as coating of gold surfaces or surfaces of other shapes than commercially available and it is highly durable to be used several times. The effect of PNIPAm-coating on ligamentocytes has not been reported but in the case of PGE we know now that it does not have any major effect on their expression profile.
We elaborated the advantages in Lines 557-567 of the discussion section: Taken together, the stable expression profile of ligament–related components in the scaffolds seeded with sheets can be highlighted here. The advantage compared to commercially available thermoresponsive surfaces is that the novel PGE-coating does not affect cell adhesion. An interference of the coating with mouse fibroblast adhesion was reported for PNIPAm-coated dishes by Becherer et al., (2015) and for smooth muscle fibroblasts by (Stöbener et al., 2018). Compared to PNIPAm-coated dishes with a detachment time of around 30 min (Mitani et al., 2014) the detachment time on PGE-coated surface is substantially shorter (5 min). The preparation of PGE-coatings is very easy and can be done by the researchers without sophisticated equipment at low costs. The only devices required include only pipettes, water, ethanol and a simple ultraviolet light lamp.
Material and Methods
- 2.6. Authors should revise the term “Vitality” to “Viability”, as well as, throughout the manuscript text.
Response: We changed it accordingly throughout the whole manuscript.
- Authors should include gene expression sections (2.8, 2.9) first than the immunofluorescence analysis (2.7), as the gene expression profile comes first than the protein expression (as authors described in results section).
Response: We thank the reviewer for this good advice and transferred the paragraphs and the respective tables accordingly.
Results
- 3.1. In figure 1, the control “co” should be identified as “uncoated” and the “PGE” as “PGE-coated”.
Response: The abbreviation have been omitted now in the graph and hence, the legends have been revised accordingly.
- 3.2. “Observation at the timepoint of detachment (day 0), which means after 24 h culturing on the plate and immediately after detachment revealed a high cell density with many densely packed cell nuclei and some ECM. Sheets showed a random and dense cell arrangement with no differences between sheets from uncoated and coated surfaces (Fig. 2A1-D2).” I believe that the images described are Fig. 2A1-D1.
Response: That`s true we corrected it and we transferred the sentence referring to native ligament staining after that to bring the figure citations in a correct order.
- 3.2. Please correct: “AB staining revealed a faint blue staining in the ECM between cells which proved sGAG deposition already at day 0 but more pronounced at day 7 (Fig. 2C1-D2).”
Response: We referred to the correct subfigure now.
- 3.3. As in the materials and methods section, authors should replace “vitality” by “viability”
Response: The term has been substituted throughout the whole manuscript.
- 3.3. In figure 3, authors should also replace “co” by “uncoated” and “PGE” by “PGE-coated”. As for “ML”, “Monolayer” would be preferable.
Response: Done.
- It would make more sense if the information regarding from cell viability (3.3.) come first than that from cell distribution and ECM formation obtained from HE staining and AB (3.2). This is also applicable for material and methods section.
Response: We changed the order as proposed in both sections.
- 3.4. In figure 4, authors should also replace “co” by “uncoated” and “PGE” by “PGE-coated”.
Response: Done.
- 3.4. Authors should provide a better explanation about the “original/initial growth area of the 12-well plate”. Apparently, the wells were also coated and uncoated with PGE. Explain this either. I do not have access to graphical abstract.
Response: The graphical abstract was shown in the beginning of the manuscript, however, referring to that makes no sense and hence, we deleted the citation of the graphical abstract. We added instead: „for the surface of the 12 wells“ (3.4., line 365).
- 3.4. In figure 4A authors should also include images of the remaining conditions, uncoated and original growth area. The same rational for figure 4C, and authors should include the F-actin staining protocol in the material and methods.
Response: Done. We included macroscopical images of the uncoated plates at 0 and 7 days as well as an F-actin staining combined with α-smooth muscle actin immunolabeling of sheets from both surfaces and the monolayer condition in the method section now (section 2.9.). We added the respective results in the result section and discussed them now (discussion section).
- 3.5. In figure 5, authors should also replace “co” by “uncoated”, “PGE” by “PGE-coated”, and “ML” by “Monolayer”.
Response: Done.
- 3.5. “At both observation time points (day 0, immediately after sheet detachment) and day 7 the gene expression for COL1A1, DCN, TNC and MKX did not significantly differ between sheets released from PGE-coated or uncoated surfaces and monolayer (Fig. 5A-D).” This statement is not consistent to the observations from figure 5, and also to the text that followed.
Response: The statement is so far correct that there were no significant differences between both sheet types. It means, when comparing directly both sheet types, there is no significant difference in gene expressions of all genes investigated. However, significant differences between sheets and monolayers could be detected and hence, the sentence was corrected.
- 3.5. Please rephrase: “TNC gene expression was only lower in the 7-day-old cell sheets released from the uncoated plates and that of MKX was lower in those from the PGE-coated plates, both compared to the monolayers of day 0 (Fig. 5C, D).” The sentence is quite confusing.
Response: This sentence was separated in two sentences to clarify it (lines 407-409).
- 3.6. “There was no decrease in viability of cells colonizing the scaffolds. When the viability of cells in the monolayer, cells on the scaffold and those in harvested and cultured cell sheets from PGE-coated plates were compared between 0, 7 and 14 d no significant difference could be detected (Fig. 6B).” Cell viability was detected by live/dead staining, and for that reason authors should consider to standardize the term to “viability” and not “vitality” as described in other sections.
Response: Throughout the manuscript only the term „viability“ is used now.
- 3.6. Why did authors use the monolayer cultures as control for comparing with PGE-coated and the scaffolds and not also the uncoated condition as in the previous sections?
Response: Since there was no major difference in gene expression in separate sheets from PGE-coated and uncoated plates and the release was more reliable with coated plates. Hence, we decided to focus on sheets from PGE-coated surfaces for scaffold seeding. Cell sheet formation needed high cell numbers, therefore, it would be difficult to include both sheet types together with the scaffolds seeded with both sheet types. We wanted to use cells from only one donor for each independent experiment and the larger expansion time would require to work with higher passage derived cells then.
The monolayer was used as the most commonly used culture type being a 2D culture in contrast to the 3D cultures represented by sheet and scaffold cultures. Hence, it allows a relation to other publications. It was already our reference in the previous set of experiments (Figure 5). Some phenotypic instability is expected in long term monolayer cultures.
- 3.6. In figure 6C, authors should also replace “PGE” by “PGE-coated”, “ML” by “Monolayer”, and “S” by “Scaffold”
Response: Done.
- 3.7. Once again the uncoated condition was excluded. My question goes also for the fact that in previous sections was observed that no significant improvements were observed after coating the surfaces with PGE. In fact, globally the uncoated condition showed better results than that observed for the PGE-coated obtained cell sheets.
Response: Looking at Figure 5 I would disagree with the statement that uncoated conditions show better results – there were no significant differences between both sheet conditions in regard to gene expression, but more often significantly lesser gene expressions of ligament-related genes in the uncoated sheets compared to the monolayer conditions. Most important is the fact that harvesting scaffolds is much more difficult on the uncoated plates. Since there was no significant difference in gene expression in separate sheets harvested from coated and uncoated plates and the release was more reliable with coated plates we decided to focus on them. Cell sheet formation needed very high cell numbers, therefore, our approaches had to be limited to one type of sheets.
- 3.7. In figure 7, authors should use more thick lines to indicate the statistical differences. The same for figure 5.
Response: Done.
Discussion
- I agree that the PGE-coated surfaces induce a more stable detachment of cell sheets than the uncoated surfaces, and authors highlighted this as a major advantage of the system. However, from histological analysis it was observed and stated by the authors that the cell sheets recovered from uncoated surfaces presented a higher ability to produce sulphated GAGs typical of these type of cells producing ECM. Authors could comment on that.
Response: We did not quantify the alcian blue staining intensity. There is some donor variability in regard to the intensity of the AB staining in the images of both sheet types. Hence, we prefer to be cautios with our conclusion concerning a comparison between sheets from coated and uncoated plates. To show that there is no major differences, we selected more representative staining of another cell donor for both sheet types now. The former image of the 7 d old sheet from the uncoated plate showed also some background staining. Therefore, the valuating statement:
„with some more intensively stained areas in the sheets from the uncoated plates“ has been removed now.
- “The use of biomechanically suitable scaffolds can overcome this limitation.” It would be interesting if authors could provide some tensile characterization tests of scaffolds, before and after the presence of cell sheets.
Response: This would be interesting. We could not do this for the revision but it would be interesting for future studies. We checked previously only the biomechanical properties of the seeded and unseeded scaffolds (Hahn et al., 2019-reference 56). This important aspect was inserted as limitation and future work package in the discussion now (lines 517-518).
56: Hahn, J.; Schulze-Tanzil, G.; Schröpfer, M.; Meyer, M.; Gögele, C.; Hoyer, M.; Spickenheuer, A.; Heinrich, G.; Breier, A. Viscoelastic Behavior of Embroidered Scaffolds for ACL Tissue Engineering Made of PLA and P(LA-CL) After In Vitro Degradation. Int J Mol Sci, 2019. 20(18).
- From gene expression profile of figure 7, the monolayer condition presented for some markers significantly higher expression than the PGE-coated and uncoated conditions, as also described by the authors in results section. This should also be discussed by the authors.
Response: In the monolayer culture cells are in a more active condition enforced to produce novel ECM since they are deprived of their ECM during passaging. We discussed the influence of cell-ECM interaction in monolayer and sheet culture now.
- Authors should better evidence the fact that the scaffolds were able to improve the expression of important markers related to ACL formation and function. No so much from protein expression, but in terms of gene expression this was quite evident, and should be better emphasized by the authors as an improved strategy for ACL tissue engineering, complementing and improving the outcomes of using cell sheets alone. Maybe the biomechanics of the scaffolds is influencing the results, and some results on that would be interesting.
Response: We thank the reviewer for this advice and revised the discussion and conclusion accordingly.
Conclusions
- This section should be significantly improved. Authors basically state that there was no advantages of coating the surfaces with PGE, which was one of the main purposed of the study. Also, the improvements achieved with scaffolds should be mentioned or better highlighted. Authors, should mention the type of scaffolds used, and at least refer the PGE-coating vs. uncoating main achievements.
Response: Done (lines 582-559.

Reviewer 3 Report
You studied applicability of thermoresponsive PGE coating for cruciate ligamentocyte sheet formation and its influence on ligament phenotype during sheet-mediated colonization of embroidered scaffolds. This result may improve and be applied for ACL tissue engineering. The results are very good with very excellent images. This PGE coating can be very useful for improving ACL tissue engineering.
So I highly evaulate your work.
Thank you.
Author Response
Dear Editor, 27th March 2021
The authors would like to thank the reviewer again for carefully reading the manuscript and very valuable comments. We modified the manuscript according to the reviewer suggestions with a list of changes shown below. We added some novel data and citation to address the reviewer comments. All changes performed are indicated in red and underlined in the revised version of the manuscript. We hope you will find this manuscript suitable for publication in “Cells”. Please do not hesitate to contact me anytime for questions regarding this manuscript.
Sincerely,
Univ.-Prof. Dr. Gundula Schulze-Tanzil
(corresponding author)
Reviewer 3
You studied applicability of thermoresponsive PGE coating for cruciate ligamentocyte sheet formation and its influence on ligament phenotype during sheet-mediated colonization of embroidered scaffolds. This result may improve and be applied for ACL tissue engineering. The results are very good with very excellent images. This PGE coating can be very useful for improving ACL tissue engineering.
So I highly evaulate your work.
Response: Thank you very much for the encouraging comment.

Round 2
Reviewer 2 Report
Reviewer comments:
This paper describes a study in which ligamentocytes cell sheets are produced in PGE-coated surfaces and compared to the same uncoated. Moreover, the potential of functionalized PLA/(P(LA-CL)) scaffolds seeded with the cell sheets was also screened. This work is of scientific relevance, since a new type of cell sheets based on ligamentocytes is proposed and analyzed when combined with scaffolds for ACL tissue engineering. Authors addressed all my suggestions and respond to comments accordingly. Still, an adequate English revision is required.
Introduction
Q. I would keep just “introduction”. Is not usual to use subheadings in the introduction section.
Results
Q. 3.7. Once again the uncoated condition was excluded. My question goes also for the fact that in previous sections was observed that no significant improvements were observed after coating the surfaces with PGE. In fact, globally the uncoated condition showed better results than that observed for the PGE-coated obtained cell sheets.
Response: Looking at Figure 5 I would disagree with the statement that uncoated conditions show better results – there were no significant differences between both sheet conditions in regard to gene expres-sion, but more often significantly lesser gene expressions of ligament-related genes in the uncoated sheets compared to the monolayer conditions.
- I agree that the uncoated condition show no statistically significant better results than the PGE-coated condition in figure 5, but the purpose of the study was to shown that the PEG-condition was better than the uncoated condition, and from the obtained results the only advantage of the PEG-coating is for the harvesting. Also, the comparison of uncoated with monolayer is not the real purpose of the study.
Conclusions
Q. “A very rapid sheet production within 24 h was possible and the detachment required in mean only 5 min.” Authors should indicate the condition that allowed this to happen: “Cell culture plates were coated with PGE, allowing a very rapid sheet production within 24 h was possible and the detachment required in mean only 5 min”
Rephrase: “The PGE-coating did not significantly affect viability and gene expression profile when compared to sheets harvested from uncoated plates, however the cell-sheets harvesting was significantly improved”.
Replace: “separate sheets” by “cell sheets alone”
Rephrase as example: “Thus, the cell sheet-seeded P(LA-CL)/PLA scaffolds has shown to be suitable for ACL tissue engineering, improving the biological outcomes of using cell sheets alone”.
Author Response
Reviewer comments:
This paper describes a study in which ligamentocytes cell sheets are produced in PGE-coated surfaces and compared to the same uncoated. Moreover, the potential of functionalized PLA/(P(LA-CL)) scaffolds seeded with the cell sheets was also screened. This work is of scientific relevance, since a new type of cell sheets based on ligamentocytes is proposed and analyzed when combined with scaffolds for ACL tissue engineering. Authors addressed all my suggestions and respond to comments accordingly. Still, an adequate English revision is required.
Response: we performed an English revision
Introduction
- I would keep just “introduction”. Is not usual to use subheadings in the introduction section.
Response: we removed the subheadings in the introduction section again.
Results
- 3.7. Once again the uncoated condition was excluded. My question goes also for the fact that in previous sections was observed that no significant improvements were observed after coating the surfaces with PGE. In fact, globally the uncoated condition showed better results than that observed for the PGE-coated obtained cell sheets.
Response: It means Fig. 6. We could not gain enough primary cells to perform the total setting simultaneously and since we compared already the gene expression of cells harvested from coated and uncoated surfaces in figure 5, we decided to focus on PGE-coated surfaces. These surfaces bear the advantage of easy sheet harvesting.
Former Response: Looking at Figure 5 I would disagree with the statement that uncoated conditions show better results – there were no significant differences between both sheet conditions in regard to gene expres-sion, but more often significantly lesser gene expressions of ligament-related genes in the uncoated sheets compared to the monolayer conditions.
- I agree that the uncoated condition show no statistically significant better results than the PGE-coated condition in figure 5, but the purpose of the study was to shown that the PEG-condition was better than the uncoated condition, and from the obtained results the only advantage of the PEG-coating is for the harvesting.
Response: We did not hypothesize or state as an aim in the manuscript that we indeed expect a „better“ performance of the PGE-coated surfaces in regard to gene expression.
Please refer to our expectations/research question stated in the abstract or summarized at the end of the introduction section:
The aim stated in the abstract is:
Lines 35-38: „we evaluated the applicability of a thermoresponsive poly(glycidyl ether) (PGE) coating for cruciate ligamentocyte sheet formation and its influence on ligamentocyte phenotype during sheet-mediated colonization of embroidered scaffolds.“
Lines 151-154: „Hence, key questions in this study are whether PGE allows the formation of sheets of primary cruciate ligamentocytes, whether the coating affects viability and ligamentocyte phenotype. Finally, it should be tested whether ligamentocyte sheets from PGE-coated surfaces could be used to seed embroidered scaffolds and if the ligament related phenotype is then maintained on the scaffolds.“
We added previously additional advantages of PGE coating (see page 22, discussion section):
The advantage of the novel PGE-coating compared to commercially available thermoresponsive surfaces is that it does not affect cell adhesion and phenotype. An interference of the commercially available PNIPAm coating with mouse fibroblast adhesion was reported previously by Becherer et al., and for smooth muscle fibroblasts by Stöbener et al., [15, 27]. Compared to PNIPAm-coated dishes with a detachment time of around 30 min [6] the detachment time on PGE-coated surface is substantially shorter (5 min). The preparation of PGE-coatings is very easy and can be done by the researcher on its own without sophisticated equipment and at low costs. The devices required include only pipettes, water, ethanol and a simple ultraviolet light lamp.
Also, the comparison of uncoated with monolayer is not the real purpose of the study.
Response: We agree, but it allows to visualize the pecularities of cell behaviour in sheets indicating that sheets represent a type of 3D culture and it allows comparison to other studies which include monolayers.
Conclusions
- “A very rapid sheet production within 24 h was possible and the detachment required in mean only 5 min.” Authors should indicate the condition that allowed this to happen: “Cell culture plates were coated with PGE, allowing a very rapid sheet production within 24 h was possible and the detachment required in mean only 5 min”
Response: We rephrased this sentence in the conclusion section as proposed (lines 582-583): „Coating of cell culture plates with PGE allowed a very rapid sheet production within 24 h and the detachment required in mean only 5 min.“
Rephrase: “The PGE-coating did not significantly affect viability and gene expression profile when compared to sheets harvested from uncoated plates, however the cell-sheets harvesting was significantly improved”.
Response: We thank the reviewer for this good advice and rewrote this sentence: (line 585-586) „Viability and gene expression profile of ligamentocytes in sheets harvested from PGE-coated plates did not significantly differ from cell performance in sheets harvested from uncoated plates.“
Unfortunately we could not find the sequence: „cell-sheets harvesting was significantly improved“ in our manuscript.
Replace: “separate sheets” by “cell sheets alone”
Response: We thank the reviewer for this good advice. We changed it (line 589).
Rephrase as example: “Thus, the cell sheet-seeded P(LA-CL)/PLA scaffolds has shown to be suitable for ACL tissue engineering, improving the biological outcomes of using cell sheets alone”.
Response: we are grateful for the proposed sequence – it included a novel aspect - and added it at the end of the discussion section.
